# High-Voltage LC-Parallel Resonant Converter with Current Control to Detect Metal Pollutants in Water through Glow-Discharge Plasma

**Pedro J. Villegas** [1,*][image: ID], **Daniel González Castro** [1], **Juan A. Martínez-Esteban** [1], **David Blanco Fernández** [2][image: ID], **Germán Marcos-Robredo** [3] and **Juan A. Martín-Ramos** [1]

1   Area Tecnología Electrónica, University of Oviedo, 33204 Gijón, Spain; uo198344@uniovi.es (D.G.C.); jamartinez@uniovi.es (J.A.M.-E.); jamartin@uniovi.es (J.A.M.-R.)
2   Area de Ingeniería de los Procesos de Fabricación, University of Oviedo, 33204 Gijón, Spain; dbf@uniovi.es
3   Area de Máquinas y Motores Térmicos, University of Oviedo, 33204 Gijón, Spain; marcosgerman@uniovi.es
*   Correspondence: pedroj@uniovi.es

**Abstract:** This paper presents a high-voltage power source to produce glow-discharge plasma in the frame of a specific application. The load has two well-differentiated types of behavior. To start the system, it is necessary to apply a high voltage, up to 15 kV, to produce air-dielectric breakdown. Before that, the output current is zero. Contrarily, under steady state, the output voltage is smaller (a few hundred volts) while the load requires current-source behavior to maintain a constant glow in the plasma. The amount of current must be selectable by the operator in the range 50–180 mA. Therefore, very different voltage gains are required, and they cannot be easily attained by a single power stage. This work describes why the LC-parallel resonant topology is a good single stage alternative to solve the problem, and shows how to make the design. The step-up transformer is the key component of the converter. It provides galvanic isolation and adapts the voltage gain to the most favorable region of the LC topology, but it also introduces non-avoidable reactive components for the resonant net, determining their shape and, to some extent, their magnitude. In the paper, the transformer's constructive details receive special attention, with discussion of its model. The experimental dynamic tests, carried out to design the control, show load behavior that resembles negative resistance. This fact makes any control loop prone to instability. To compensate this effect, a resistive ballast is proposed, eliminating its impact on efficiency with a novel filter design, based on an inductor, connected in series with the load beyond the voltage-clamping capacitor. The analysis includes a mathematical model of the filtering capacitor discharge through the inductor during the breakdown transient. The model provides insight into the dimensions of the inductor, to limit the discharge current peak and to analyze the overall performance on steady state. Another detail addressed is the balance among total weight, efficiency and autonomy, which appears if the filter inductor is substituted for a larger battery in autonomous operation. Finally, a comprehensive set of experimental results on the real load illustrate the performance of the power source, showing waveforms at breakdown and at steady state (for different output currents). Additionally, the detector's constructive principles are described and its experimental performance is explored, showing results with two different types of metallic pollutants in water.

**Keywords:** environmental engineering; water pollution; glow-discharge devices; DC–DC power converters; resonant inverters; high-voltage techniques; current control

## 1. Introduction

Human activity involves handling large quantities of materials which have always been buried away in the earth's crust. There is a real risk of part of these materials, heavy metals among them, filtering to the environment, entering in the food chain and affecting the ecosystems [1]. In this context, pollution of hydric resources is particularly worrying.

Of course, precise laboratory techniques have been developed to measure the presence of heavy metals in water [2–7], but the use of plasma through air at atmospheric pressure has only been proposed recently [8–11]. In this technique, the absence of a vacuum chamber makes possible a portable device. This paper analyses how to power an original portable device capable of measuring the presence of water pollutants in situ. Figure 1 shows the functional blocks of such equipment: The water being tested is pumped through a hollow, grounded electrode, running over it. Between 3 mm and 5 mm away, there is a second electrode, connected to a positive voltage. Air at atmospheric pressure is between them. To start the system out, the power source increases the voltage to cause the electrical breakdown of the air in the gap. Next, it regulates the current to establish a stable glow-discharge between the electrodes. The interaction between the liquid and the plasma transfers traces of the materials present in the water to the center of the glow discharge. A miniaturized optical emission spectroscope is optically coupled to detect the characteristic spectra of metals. Data from the spectrometer are transferred to a computer where they are treated by means of software.

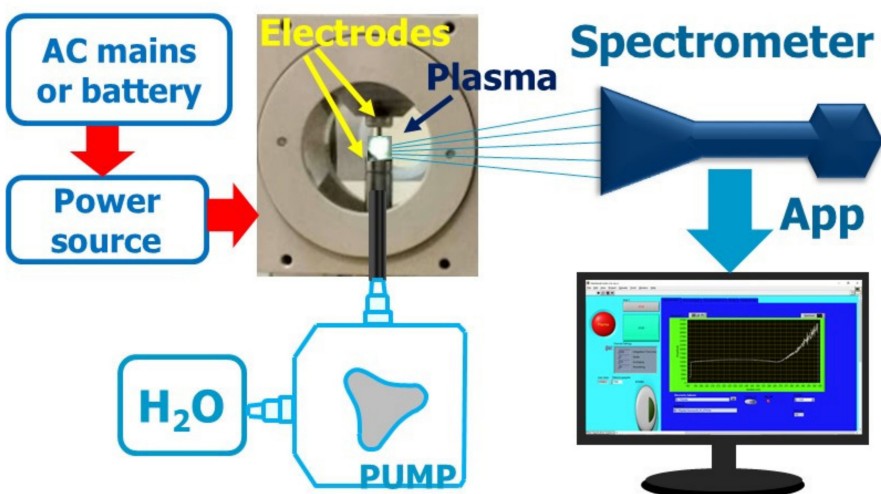

**Figure 1.** Conceptual scheme of the portable pollutant detector.

To provide portability to the equipment, the primary energy source is a 48 V battery, which is the converter input voltage. However, a 150 W AC/DC adaptor supports the operation from the AC mains.

On the other hand, at the output, the power source must operate in two very different conditions:

1. Ignition: To establish glow discharge between electrodes, it is necessary to break the dielectric strength of the air in the gap first. In atmospheric air, under the worst conditions, an electric field of 3 kV/mm is enough for this purpose. Therefore, the power source must be capable of generating up to 15 kV. However, the actual value will depend on the operating conditions, such as the atmospheric pressure, the air moisture content or the configuration of the electrodes. To reach the required voltage level, the capacitor at the output of the power source is charged by the resonant stage. Voltage and current conditions change dynamically throughout the circuit in this charging process. It might be considered as a transient, which would end in a high output voltage and null output current state.

2. Glow discharge: Just after dielectric breakdown, the output voltage must diminish instantaneously to a considerably lower value between 450 V to 750 V DC, to establish normal glow through the gap. Now, a stable operation requires controlling the output current instead the voltage. The operator has the chance to select the reference in a range from 70 mA to 180 mA DC.

These two operating conditions force a wide variation in voltage gain, $G_V$. It must be above 300 to ensure dielectric breakdown (1), and it may be below 10 when at normal glow (2).

$$Max.\ volt.\ gain = G_V^{Max} = \frac{15\ \text{kV}}{48\ \text{V}} = 312.5 \tag{1}$$

$$Min.\ volt.\ gain = G_V^{Min} = \frac{450\ \text{V}}{48\ \text{V}} \simeq 9.4 \tag{2}$$

$$R = \frac{G_V^{Max}}{G_V^{Min}} = \frac{312.5}{9.4} = 33.33 \tag{3}$$

A very common approach in other plasma applications is the use of several stages in a cascade [12,13]. However, for this low power, portable application of the final structure should be as simple as possible. Bearing this in mind, resonant converters have the capability to adapt to very different output conditions, and they are widespread in the industry for many applications [14–18]. In fact, a comprehensive review of power supplies for plasma material processing [13] shows that solutions based on LLC or LCC resonant structures have already been proposed as intermediary power conversion systems [19–21]. In this paper, however, the LC-parallel resonant topology is preferred. It presents a simpler resonant net and can integrate easily the step-up transformer and its parasitic components. Moreover, the application requires a wide variation in the gain (3), which requires a power topology capable to deal inherently with very different conversion ratios. The LC solution presents such a flexibility to a greater degree than other resonant topologies and their features match perfectly with those of the application: at no load resonance leads to high-voltage gain, while voltage attenuation is possible at different output currents without a large excursion in switching frequency. Therefore, the LC resonant structure is a better candidate for the application. In the next paragraph, the model of the LC-parallel resonant topology with a capacitor as the output filter also known as PRC-C (Figure 2) is used to carry out the design.

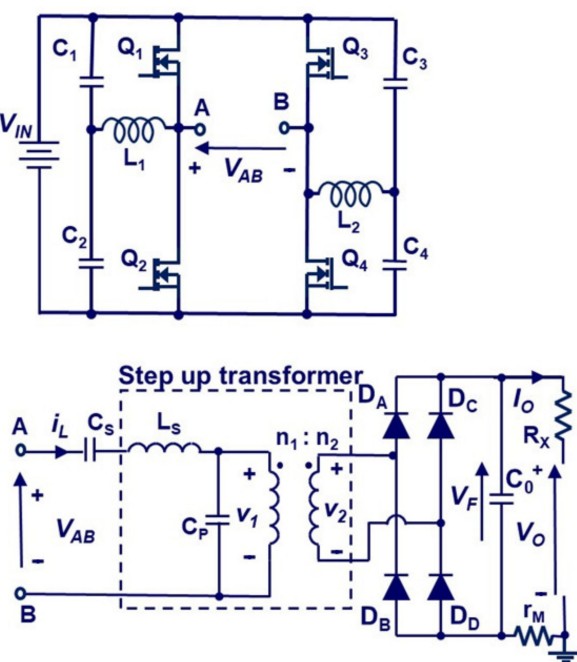

**Figure 2.** Parallel resonant converter with a capacitive output filter, PRC-C. An additional passive network, ($L_1$, $L_2$, $C_1$–$C_4$) is added to obtain soft switching in the whole inverter operating range.

## 2. Power Source Design

Frequency, f, and duty cycle, d, are the control parameters in PRC-C topology [22,23]. Different combinations of them lead to very different values in intrinsic voltage gain, $IG_V$ (the total gain excluding transformer contribution). Under no load condition it is possible to obtain $IG_V = 10$ if the converter operates near resonance, with a large d, and the quality factor of the net is high enough. To be conservative, this design considers a maximum $IG_V \simeq 7.5$, resulting in a transformer ratio of $n_1$:$n_2$ equal to 1:42 (4).

$$\frac{V_O^{Max}}{V_{IN}} = IG_V^{Max} \cdot r_T \rightarrow \frac{15\text{kV}}{48} = 7.5 \cdot r_T \rightarrow r_T = 42 \tag{4}$$

For most of the converters, 7.5 represents a non-reachable intrinsic gain. Even for PRC-C, it requires a careful design, including the step-up transformer. However, it is not advisable to lower it because that would increase $r_T$, and a larger $r_T$ would be a drawback when a smaller output voltage is needed. In fact, most of the time, the power source will operate in the glow-discharge region of 400 V–600 V. This means an $IG_V$ ranges from approx. 0.2 to 0.35 if $r_T$ is still equal to 42. Globally, $IG_V^{Max} = 7.5$ and $IG_V^{Min} = 0.2$ is challenging for any topology, and not many show the versatility of the PRC-C.

PRC-C topology also has the ability to cope with the step-up transformer non-idealities. The secondary winding must withstand 15 kV, allowing large isolation distances from the primary winding and the core, which are grounded. These distances cause a sizable value of leakage flux, modelled by a series inductance, $L_S$, which cannot be neglected. PRC-C can include this in the resonant net. At the same time, in any winding, a small capacitive effect [22–25] appears between any two adjacent turns, and there are many in the secondary winding of this step-up transformer ($r_T$ = 42). Therefore, a noticeable capacitive effect can be measured in the terminals. In Figure 1, this is modelled by a parallel capacitor which has been transferred to the primary, $C_P$. Again, PRC-C welcomes this capacitance in the parallel resonant net.

In the experimental transformer (Figure 3) the secondary winding occupies three layers, which have been separated in three concentric coils, ensuring isolation and lowering tolerances in the parasitic components. Table 1 shows the experimental transformer parameters when measured with an impedance analyzer. $L_M$ represents the magnetizing inductance.

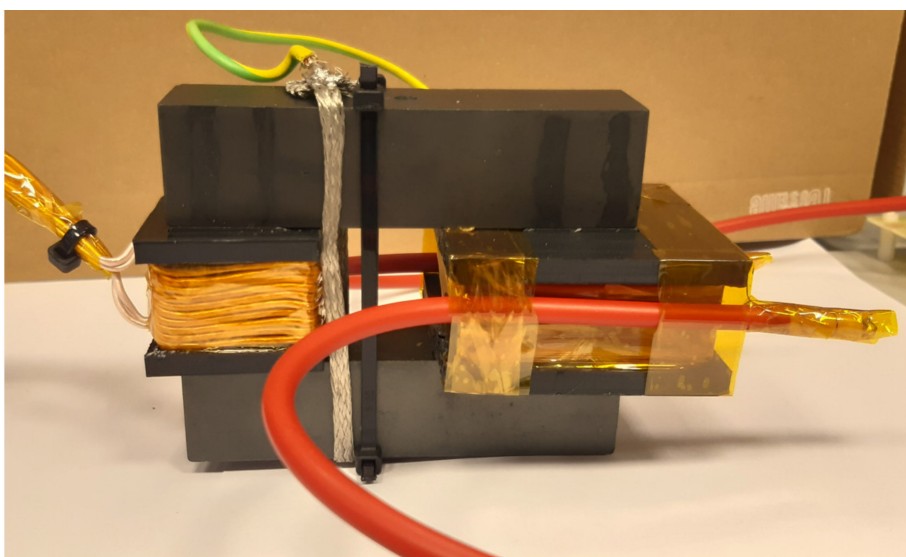

**Figure 3.** Step up transformer: left: primary winding; right: concentric high-voltage coils.

**Table 1.** Experimental parameters for the wound step-up transformer.

| $L_S$ | $L_M$ | $C_P$ | $n_1$ | $n_1{:}n_2$ |
|---|---|---|---|---|
| 8.3 µH | 96 µH | 17.5 nF | 5 | 1:42 |

In the resonant stage, switching and resonant frequency are similar. However, according to the values of Table 1, the resonance frequency for $L_S$ and $C_P$ is too high. To limit switching losses, an external capacitor, 340 nF (measured from primary), is connected in parallel to the secondary winding, diminishing $f_{RES}$ to a more favorable value of 75 kHz (5). On the other hand, the series capacitor, $C_S$, does not participate in the resonant net. It just blocks any remaining DC voltage asymmetry in $V_{AB}$.

According to Table I, the resonant frequency of $L_S$ and $C_P$ is too high. To limit switching losses, an external capacitor, 344 nF (measured from primary), is connected in parallel to the secondary winding, diminishing $f_{RES}$ to about 90 kHz. On the other hand, the series capacitor in Figure 2, $C_S$, does not participate in the resonant net. It just blocks any residual DC level coming from residual asymmetries in $V_{AB}$.

For comparison purposes, another experimental transformer is assembled using an alternative secondary winding made with PCBs separated by cardboard. The major difference between prototypes is a lower leakage inductance which reduces resonance to 75 kHz (5). Except for that detail, the behavior of the entire converter is similar in terms of waveforms and efficiency.

$$f_{RES} = \frac{1}{2 \cdot \pi \sqrt{L_S \cdot C_P}} = \frac{1}{2 \cdot \pi \sqrt{11 \cdot 10^{-6} \cdot 391 \cdot 10^{-9}}} = 75 \text{ kHz} \tag{5}$$

As a first approach, the output filter was formed by a single capacitor. In this way, the filter is simple, but the output rectifier works in discontinuous conduction mode, affecting the resonance between $C_P$-$L_S$ [26]. As Figure 2 shows, the rectifier is the only element separating $C_P$ and $C_0$, being $C_0 \gg C_P$, and it will be off whenever $C_P$ voltage is low. In this part of the period, the resonant current, $i_L$, will go through $C_P$, charging it. Once $C_P$ voltage tries to exceed $V_F$ (reflected to primary), the rectifier connects $C_0$ in parallel, clamping the $C_P$ voltage. Hence, the voltage in the secondary winding is never higher than the breakdown voltage, fixing the isolation requirements.

A small resistor, $r_M$, provides a measurement of the output current, which must be regulated. This resistor is ground connected as the positive electrode, providing a negative voltage. The regulation is implemented through a current-mode control which programs the duty cycle in the inverter, d, while the switching frequency, $f_S$, is initially constant. For high output currents, such as 175 mA, the expected d is also high, d = 0.4, approx. Then, the inductive behavior of the resonant net is enough to ensure soft switching in the inverter (see experimental results). However, if the target current diminishes, the duty cycle must also diminish, and the current phase may not be inductive enough to maintain soft switching. To solve the problem, further inductivity is provided by increasing switching frequency once the current reference has diminished below a threshold. Figure 4 shows how the switching frequency is maintained in resonance when the output current target is above 90 mA. Below that limit, the reduction in the duty cycle is stopped, and the regulation is made by frequency variation.

The output filter of the topology, where there is only a capacitor, affects the resonance $C_P$-$L_S$. In fact, avoiding inductors in the high-voltage side, which would be bulky because of their isolation requirements, makes the filter simpler, but causes the output rectifier to work in discontinuous conduction mode [24]. In fact, the rectifier separates $C_P$ and $C_0$, with $C_0$ having several times the capacitance of $C_P$. Therefore, resonant current, $i_L$, charges $C_P$ while the rectifier is off. Once $C_P$ voltage tries to exceed $V_F$ (reflected to primary), $C_0$ is connected in parallel, clamping $C_P$ voltage. Hence, voltage in the secondary winding is never higher than the breakdown voltage, fixing isolation requirements.

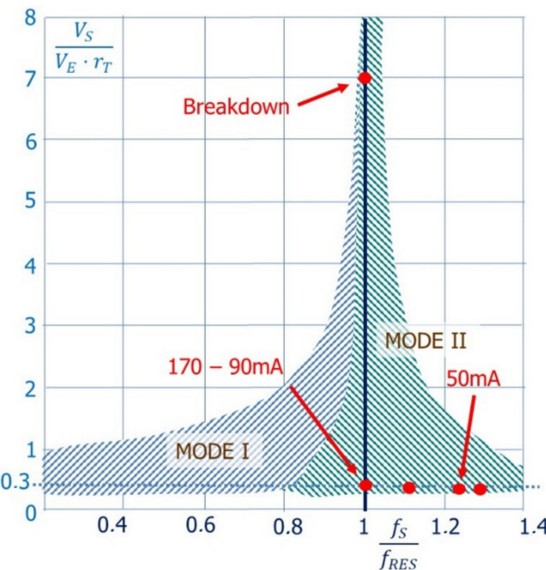

**Figure 4.** Voltage gain versus normalized switching frequency and the position of some operation points.

Regulation of the output current is one of the application requirements. A small resistor, $r_M$, in series with the load, allows current measurement. This resistor is ground connected as the positive electrode and provides a negative voltage. A current mode control uses it to program the inverter duty cycle, d, while switching frequency, $f_S$, is initially constant. For high-output currents, such as 175 mA, the expected d is high at approx. d = 0.4. Under these conditions, the inductive behavior of the resonant tank is enough to ensure soft switching (see experimental results). However, if the current reference diminishes, duty cycle diminishes accordingly, and the resonant current phase may not be inductive enough to maintain this soft switching. To solve the problem, further inductivity is provided by increasing switching frequency once the current reference has diminished below a threshold (Figure 3). At the same time, two small inductors, $L_1$ and $L_2$, have been added to the topology to provide an inductive current throughout the working area. This reactive net will assist in the charging–discharging switches' parasitic capacitances. Figure 5 shows a picture of the inverter in a prototype. Inductors $L_1$ and $L_2$ present an inductance of 27 μH and are made using a RM10 core. Their losses are limited to 1 W in the worst conditions.

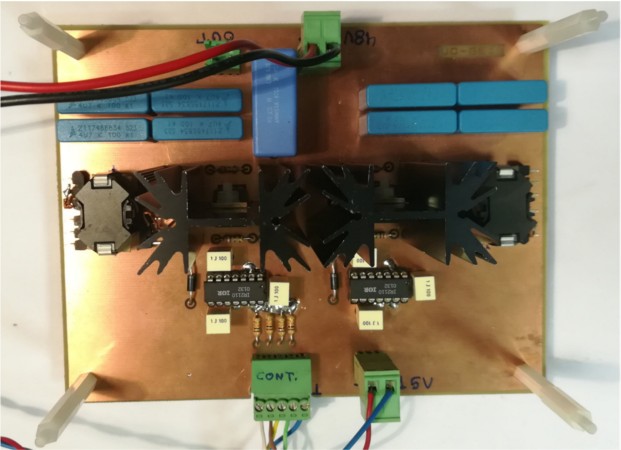

**Figure 5.** Inverter of a prototype. RM10 snubber inductors appear on the borders. PCB size has not been optimized.

## 3. Dynamic Behavior

The output capacitor, $C_0$, stores a certain amount of energy when charged to the break-down voltage, which must be quickly removed when conduction is established, through the gap between electrodes. The output voltage must evolve very fast from several kV to about 400 V. If the capacitor were connected in parallel with the gap, it would produce a high discharge current which would be unlikely to make a successful glow discharge. Adding a resistor, $R_X$, in series with the gap, as in Figure 2, is a simple way to limit the current peak in the transition. However, it would damage converter efficiency in normal operation.

On the other hand, the control of the output current requires knowledge of the system dynamic response, the load and the topology together, to design the feedback loop. This behavior can be experimentally measured with an online digital system [27,28], which adds a slight perturbation to the inverter duty cycle when the power source is working at steady state. Such a perturbation has a constant magnitude and a sinusoidal profile, whose frequency is varied from 10 Hz to half of the switching frequency at regular steps. Both, plasma current and voltage are recorded in magnitude and phase for every step, leading to small signal dynamic plots of the system. Those plots relate the control parameter, duty cycle, against the behavior of the output for different frequencies. However, as an example, Figure 6 shows—in magnitude, left, and phase, right—the low-signal experimental impedance of the load at several frequencies. Up to 1 kHz, the phase of the load is about 180°, which means negative resistance behavior of a magnitude 1 kΩ (Figure 6). The measurement shown in Figure 6 is approximately valid up to one eighth of the switching frequency, i.e., up to 12 kHz. From 1 kHz to 12 kHz, the impedance phase moves from 180° to 100° becoming a complex number, though its real part is always negative.

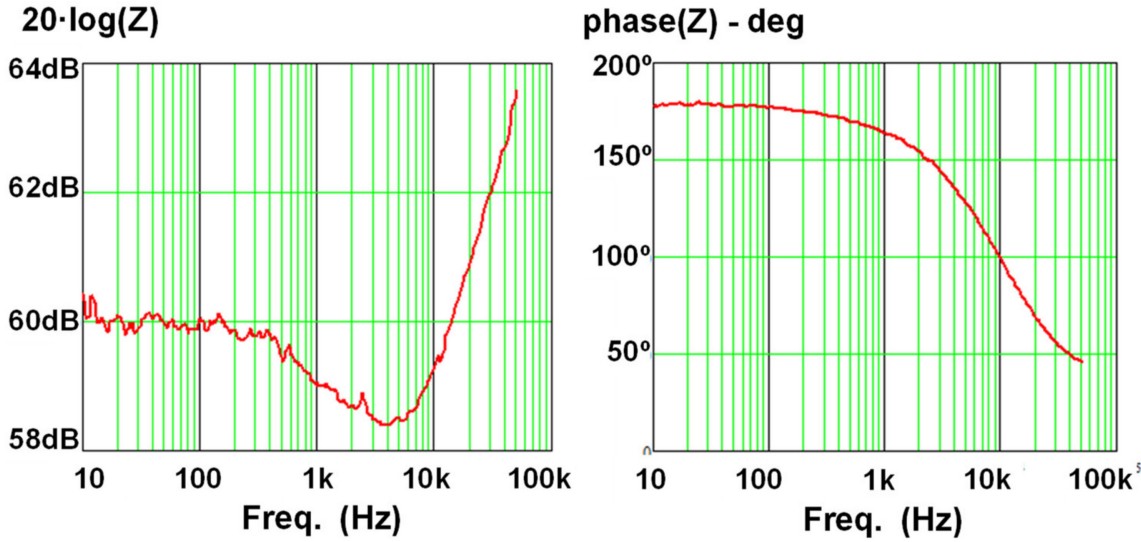

**Figure 6.** Experimental, small signal characterization of the glow-discharge impedance. This was obtained from dynamical measurements of the output voltage and current.

## 4. Experimental Tests

Figure 7 shows the behavior of the power source when the output current is 100 mA. The secondary voltage, $V_2$ (in yellow), is limited, twice in a period, to the output voltage (±700 V in this case) having resonant transitions between both values. The switching frequency is 77 kHz, near resonance, as expected. The control adjusts the duty cycle to obtain the desired current. Although the duty cycle is quite low, the resonant current, $iL$ (purple), is still lagging the inverter voltage, $V_{AB}$ (red), producing soft switching.

Figure 8 shows the other two operation points. For high current levels, 170 mA, the duty cycle is large and there is enough inductive current to support soft switching. The maximum resonant current reaches 15 A while the output voltage, to maintain the plasma,

is 438 V. However, when the output current is low, 50 mA, the only chance to maintain the lagging current is to increase the switching frequency, which is 92 kHz in the image. The inductors from the snubbers now assure soft transitions by injecting about 2 A to the charge–discharge inverter switches' parasitic capacitances. There is no voltage spike in $V_{AB}$ of Figure 8 (bottom), even though there is no resonant current available in some of the MOSFET transitions.

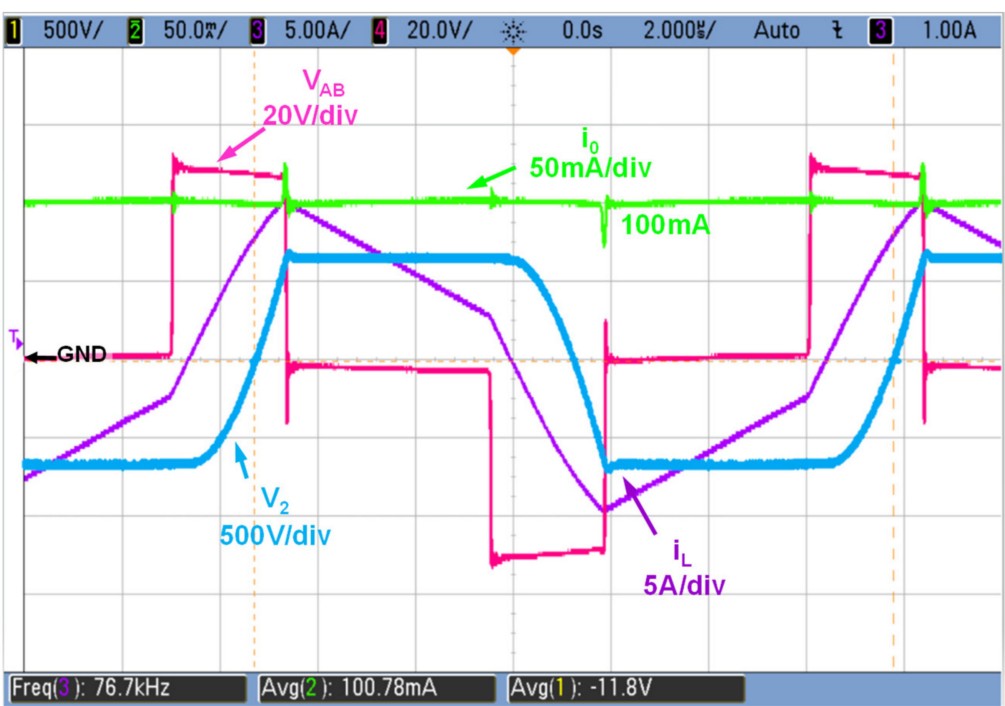

**Figure 7.** Experimental waveforms for glow discharge when the output current is 100 mA. Switching frequency is 76.7 kHz, d is 0.18 and $V_F$, not in the image, is 632 V.

The evolution of the output voltage before, during and after breakdown is recorded in Figures 9 and 10. At the top of Figure 9, the output voltage evolves from approx. 0 V to 7 kV to achieve a breakdown in the gap. The capacitor charges for about 1.25 s (500 ms/div). At breakdown, the output capacitor is quickly discharged through RL and the gap. At the bottom of Figure 9, a zoom shows the breakdown in more detail, with a time scale of 200 µs/div. In the transient, the output current stabilizes at 600 µs, later than the reference, which is 70 mA in this case.

In Figure 10, the emphasis is placed on the resonant current and voltage during breakdown. A single capture is recorded in the oscilloscope, with the zoom moved to investigate the different zones of operation. Before breakdown (Figure 10a) the output current and delivered power is null. In fact, the phase between the current, $i_L$, and voltage, $V_{AB}$, is 90°. After breakdown (Figure 10b) the waveforms are similar to those represented in Figures 8 and 9. The breakdown causes an output current spike proportional to the breakdown voltage and the limiting resistor, $R_X$, as the magenta plot shows in the upper side of Figure 10a,b.

Dynamically, the control allows changing of the output current at 1 kHz. Figure 11 shows how the reference varies at that speed from 90 mA to 110 mA, and measures the response of the actual output current when following the target.

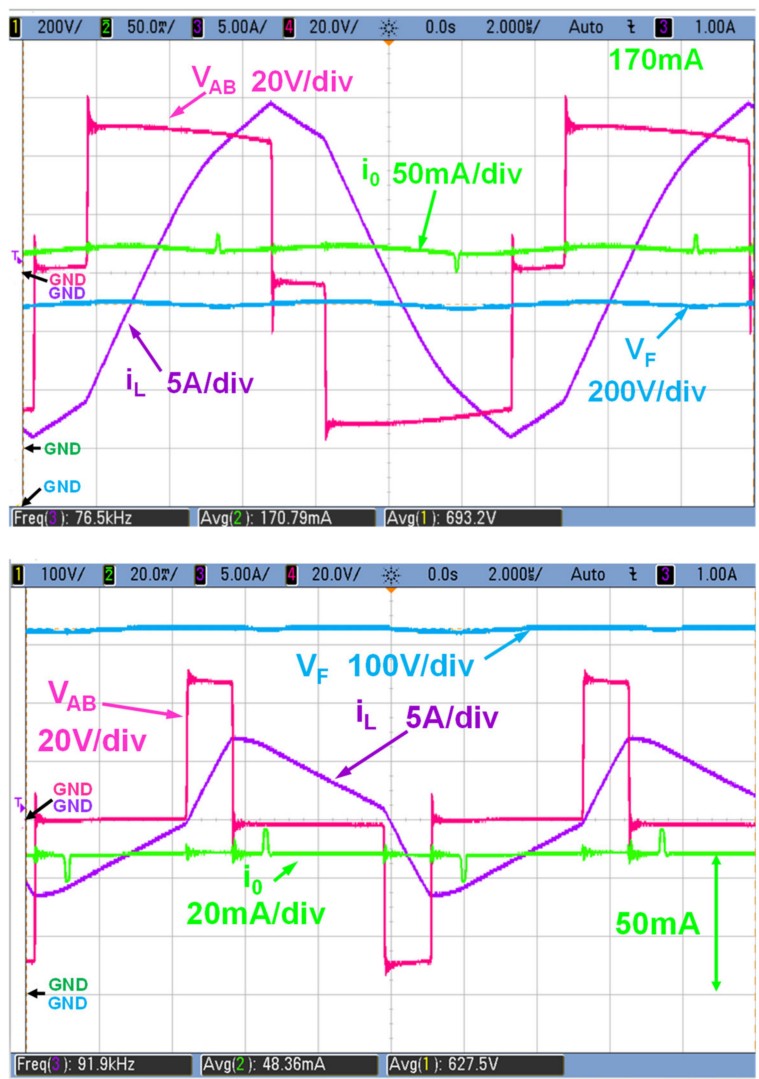

**Figure 8.** Experimental waveforms for glow discharge. Top: $f_S$ = 76.5 kHz, d = 0.39, $i_0$ = 170 mA, $V_0$ = 693 V. Bottom: $f_S$ = 91.9 kHz, d = 0.11, $i_0$ = 50 mA, $V_0$ = 627 V.

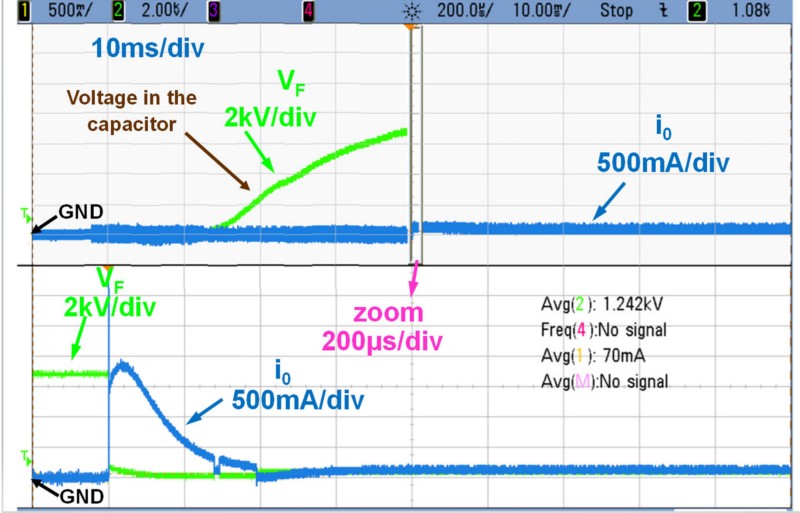

**Figure 9.** Output current, $i_0$, and capacitor voltage, $V_F$, evolution before and after breakdown when distance between electrodes is 3 mm.

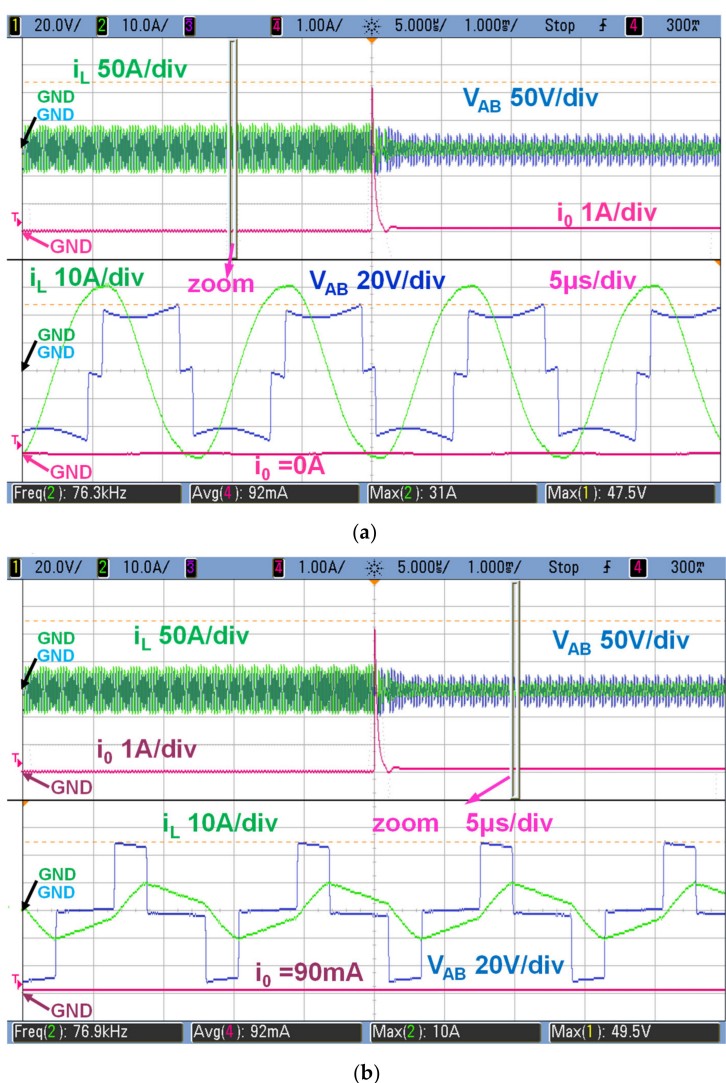

**Figure 10.** Inverter voltage, $V_{AB}$, resonant current, $i_L$, and output current, $I_0$, before (**a**), and after (**b**) breakdown.

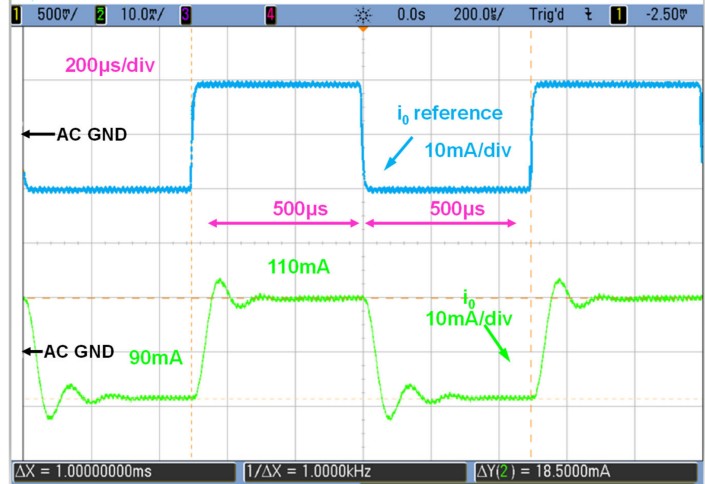

**Figure 11.** Dynamic response of the output current facing a change in the reference every 500 μs.

### 5. Autonomy of the Equipment

Although useful, $R_X$ is a major drawback for good efficiency. Considering a value of 1.5 kΩ, it dissipates 15 W at nominal conditions (100 mA) when the total output power is 47 W. However, it cannot be completely removed because a resistor is necessary to limit the current in the transition between the breakdown and glow discharge, smoothing the discharge of the output capacitor. $R_X$ also aids to stabilize the system under steady state, since the discharge behaves dynamically as a negative resistor (Figure 6). As an alternative, an inductor, LF, in parallel with $R_X$, has been introduced to improve efficiency (Figure 12). Now, $_{RX}$ is still dominant in the transients, where the inductor reacts slowly, but it is short-circuited in steady state operation due to the low impedance path of the inductor in DC operation. Consequently, the power rating of $R_X$ can be lowered and its size is reduced, at the same time as efficiency is enhanced.

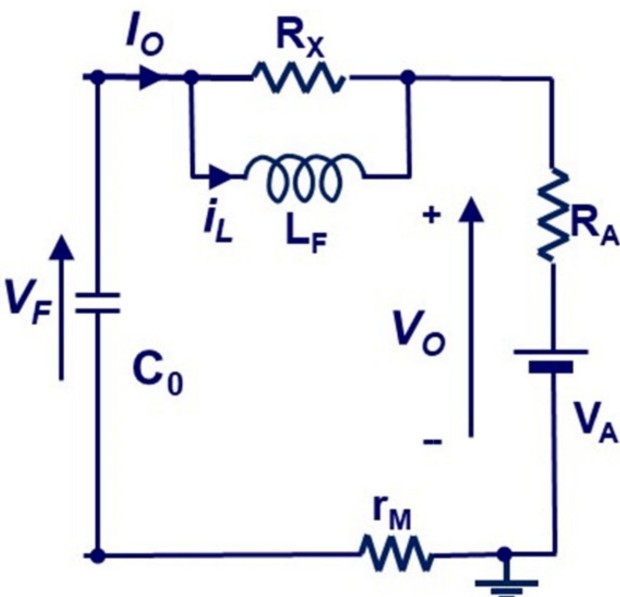

**Figure 12.** Second-order output filter to improve efficiency.

The analysis of the filter (Figure 12) in the discharge process follows Kirchhoff's laws, as given in (6), see Appendix A:

$$\left. \begin{array}{l} L_F \frac{d\,i_L}{dt} = R_X(I_0 - i_L) \\ V_F = \frac{1}{C_0} \int_{-\infty}^{t} -I_0 \cdot dt = L_F \frac{d\,i_L}{dt} + R_A \cdot I_0 + V_A \end{array} \right\} \tag{6}$$

Operating in (6), the behavior of the filter is demonstrated to be second order in terms of output and inductor current (7)–(11). See the annex.

$$\frac{d^2 i_L}{dt^2} + b\frac{di_L}{dt} + c = 0 \tag{7}$$

$$b = \frac{1}{\tau_L} + \frac{1}{\tau_C} \tag{8}$$

$$\tau_C = C_0 \cdot (R_A + R_X) \tag{9}$$

$$\tau_L = L \cdot \frac{R_A + R_X}{R_A \cdot R_X} \tag{10}$$

$$c = \frac{R_X}{L \cdot \tau_C} \tag{11}$$

Initial conditions to solve (7) are given in (12), where it is assumed that the initial current at breakdown is null and the capacitor $C_0$ is charged to the breakdown voltage of the gap, $V_{BD}$:

$$\left. \begin{array}{c} i_L(0) = 0 \\ \frac{d\,i_L}{dt}\Big|_{t=0} = \frac{V_{BD} - V_A}{R_A \cdot \tau_L} \end{array} \right\} \tag{12}$$

The design of the inductance relies on the desired maximum value of the transient current. Solving (7), it can be demonstrated that the higher the inductance, the lower the transient current through it. It is very important that the inductor does not saturate in the transient, since it would also short-circuit $R_X$ at that moment, leading to a fast discharge of the output capacitor, $C_0$, through the gap; this would transform the breakdown to an arc and prevent the formation of a glow discharge.

There is a trade-off between the value of the inductance and the peak current in the design. For instance, if the capacitor voltage at breakdown is 6 kV, a large inductor of 350 mH will have to withstand 0.64 A in the discharge without saturation (7)–(12). This current should be considered as high, since nominal conditions do not go beyond 180 mA in the output. A lower inductor, 40 mH, will lead to an 8.3 A peak in the discharge. The final size of LF will be related to the energy that it has to store in the worst conditions. In Figure 13, the maximum stored energy in the inductor is plotted against its inductance value. Above 350 mH, the stored energy is at a minimum.

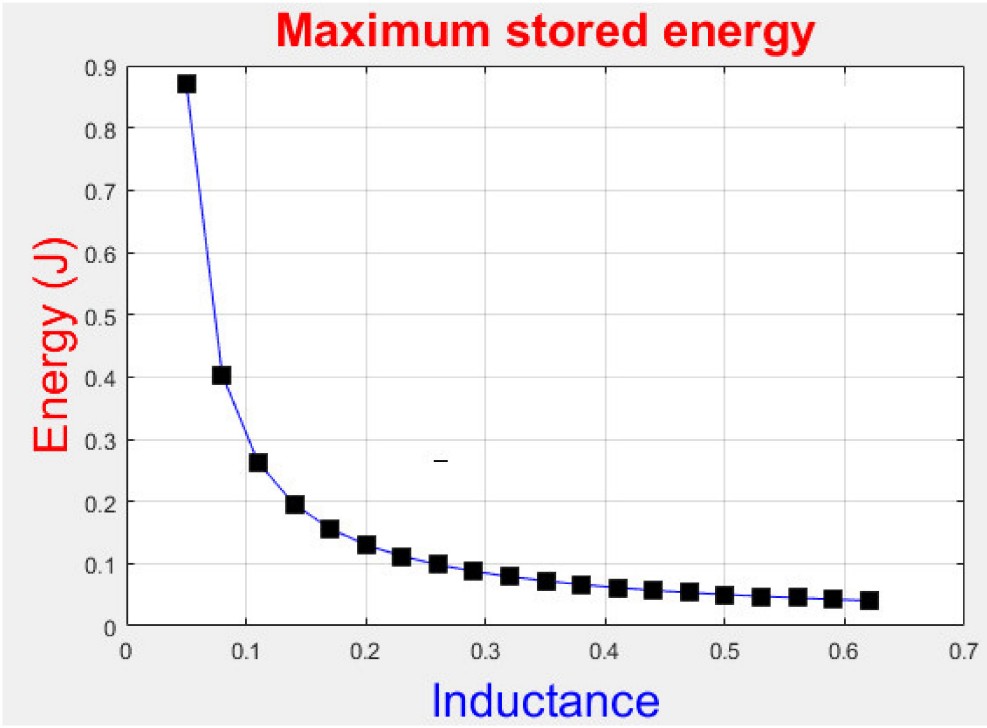

**Figure 13.** Stored energy at the inductor when the current through it reaches its maximum against inductance.

On the other hand, to ensure stability, coefficient b in (6) must be positive. Provided that RA is negative, (8) states that $R_X$ must be bigger than $R_A$, and (7) obliges $\tau_C$ to be smaller than $\tau_L$, because $\tau_L$ is always negative. Therefore, under steady-state conditions, there is a lower limit for the filtering inductor. Table 2 gives details about the values involved in the prototype filter.

**Table 2.** Values in the prototype filter.

| Measured $R_A$ | $C_0$ | Limit for $L_F$ | Selected $L_F$ | Simulated Peak Current for 8 kV | Measured Peak Current for 8 kV |
|---|---|---|---|---|---|
| 1 kΩ | 27.5 nF | >33 mH | 360 mH | 0.86 A | 0.83 A |

Figure 14 shows the experimental waveforms in the gap transition. The voltage (green) was rising in a ramp, and the breakdown happened when it reached 5.3 kV. Theoretically, the current was limited to 3.1 A by $R_X$ according to (13), close to the measured 2.9 A (7% error). On the other hand, the solution of (7) provides a theoretical maximum inductor current of 0.54 A, while the experimental measurement is 0.55 A (negligible error). Table 2 shows more data for other breakdown voltage with a different distance in the gap.

$$i_L = \frac{V_{BD} - V_{Gap}}{R_X} \sim \frac{5300 - 650}{1500} = 3.1 \ A \tag{13}$$

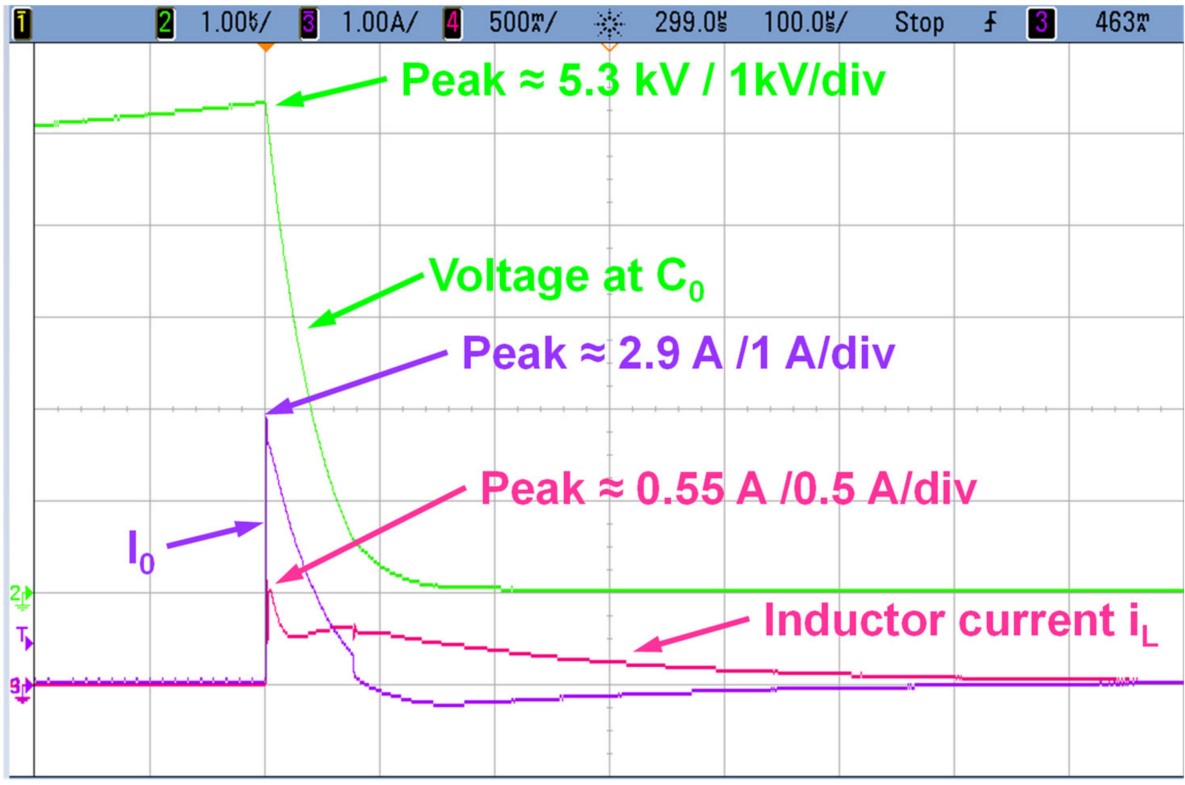

**Figure 14.** Waveforms at the output during breakdown.

An experimental filter with an inductor was assembled to validate the study. The system behaves as expected and efficiency is improved accordingly, since $R_X$ does not dissipate power in steady state anymore. The price to pay for this is the inductor weight. In fact, the mathematical study demonstrates that there is not any combination of inductance and peak current which can be materialized in a light inductor. Although it can probably be further optimized, in the experimental tests, $L_F$ weighed around 500 g, saving 15 W in nominal conditions, i.e., 15 Wh per hour. The conditions for the design are:

- The inductor must withstand the output voltage, which complicates isolation issues.
- The real inductor presents a parallel capacitance, which is instantaneously charged at breakdown. This capacitance must be maintained under control to prevent an additional starting current peak.

As an alternative, LFP batteries present power densities around 100 Wh/kg. Therefore, 500 g of inductor (15 Wh) can be compensated by 200 g extra in the battery at 80% of DoD for every hour of autonomy. The inductor is definitely better with two hours of programmed autonomy.

The rectifying diodes must be capable to withstand the high breakdown voltage. High reverse blocking capability results in a large forward voltage, which in normal operation, hinders efficiency. In the assembled prototype, approximately 5% of total output losses are caused by the output-rectifying diodes. Table 3 summarizes the projection of experimental losses and their impact on the inductor.

**Table 3.** Prototype performance.

| $I_0$ (mA) | 50 | 100 | 120 | 150 | 170 |
|---|---|---|---|---|---|
| $V_0$ (V) | 529 | 470 | 454 | 436 | 425 |
| $P_0$ (W) | 26.5 | 47 | 54.5 | 65.4 | 72.3 |
| Efficiency without $L_F$ (%) | 77 | 66 | 62 | 56 | 53 |
| Efficiency with $L_F$ (%) | 87 | 86 | 84 | 82 | 80 |

## 6. Performance of the Unit

To validate the performance of all the equipment, including the power supply, a prototype of the system is assembled and tested. In the measurements, the plasma must not present sparks or fluctuations, because they would affect the quality of the signal acquired by the spectrometer (StellarNet LHR-UV3-7). For the experiments, the length of the gap is set to 3 mm. In these conditions, the current can vary from 35 mA to 105 mA, minimum and maximum, to obtain stable plasma. Tap water is pumped though the hollow electrode at a flow rate of 2.75 cm$^3$/min. It contains an unknown quantity of Mg, which can be considered as a constant throughout the experiments. To produce the measurement, two kinds of additives are added to all of the samples:

- A pollutant, Cd, in a proportion of 0.5 ppm as the metal to be identified.
- Two acids to improve the conductivity of the solution: 0.65% of nitric acid and 1% of formic acid.

Once powered on, the system registers 600 different spectra with a total integration time of 100 ms, giving their average to the user. Those spectra were acquired directly, without optic fiber. A biconvex lens with a quartz window to avoid spatters is used instead. Figure 15 shows a summary of the experimental results from the prototype in these conditions. On the top, the intensity of the measured signal is represented against the wavelength for the particular case of 90 mA in the output current. Zooming into two different areas allows us to distinguish a peak of emission for Cd at 228.8 nm (left) and Mg at 285.2 nm (right). The net signal is defined as the difference between the maximum value of emission at the peak and the flat value at the base in the surrounding area. At the bottom of Figure 15, experimental net values are represented vs. the output current for Cd (left) and Mg (right). The higher the output current, the larger the net value obtained, with an almost linear dependence. Hence, the results with 90 mA are more representative. The net value of Mg stands out clearly in the measurements, since its concentration in the solution is quite high (several ppm) and it is an easy material to ionize. On the other hand, the concentration of Cd is low, and consequently, its net value is smaller, although distinguishable. The measurements also present other peaks distributed in the range 279–300 nm (Figure 13, top and right). They belong to the molecular band of OH.

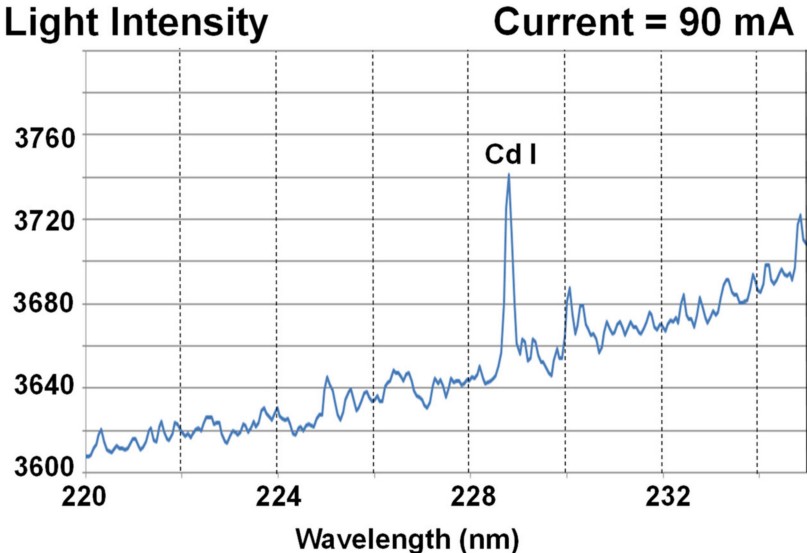

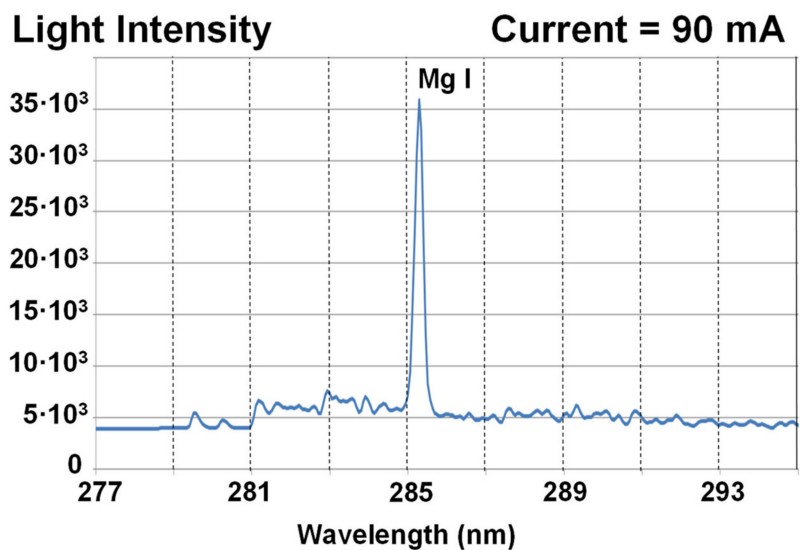

**Figure 15.** Experimental results of the system when measuring tap water containing a small quantity, 0.5 ppm, of Cd.

## 7. Discussion

In a low-power application, below 200 W, a single-stage power source is preferable. In this case, gain conditions introduce an additional difficulty since, with a wide variation ratio from 1 to 33, they are very difficult to attain. The LC-parallel resonant topology is demonstrated as a good solution to cope with both conditions. It provides more flexibility than other resonant structures, such as LLC or LCC, which have been previously proposed as intermediate stages. In fact, the parallel resonance presents the largest gain for a given quality factor, while gains well below a value of one are still possible within a short switching frequency excursion. Additionally, the LC structure adapts the parasitic components of the step-up transformer in a natural way. Regarding control, a new filter structure stabilizes overall behavior without any efficiency penalty. However, the need to use four high-voltage diodes in the output rectifier remains. Their large forward-voltage deteriorates efficiency in at least 5%, when the output voltage is low, at glowing discharge. Further improvement in efficiency might be obtained from a new rectifying structure. Overall, the final design remains simple, robust and functional.

## 8. Conclusions

This paper demonstrates that the LC-parallel resonant topology is a good candidate for a power source to maintain glowing discharge in air at atmospheric conditions. The key features of the design rely on:

1. An intrinsic voltage gain (without a step-up transformer) centered in the stage-optimum range of 0.3–7.5.
2. A step-up transformer with a magnetizing inductor that can be ignored in any resonant net and does not affect the core size. A standard size core that supports windings in separate columns with enough isolation and provides a suitable leakage inductance.

The experimental low-signal model of the load shows that its dynamic behavior is prone to instability. A new filter, based on a resistor and a filtering inductor, compensates the equivalent negative resistor of the system, not affecting the efficiency. A mathematical model of such a filter provides a guide to the design, which has been applied and experimentally demonstrated in the application. The final weight of the filter is preferable to the equivalent battery size to compensate for a worse performance. Most of remaining losses fall on the output-rectifying diodes: They present a very large forward voltage as penalty to withstand the breakdown output voltage.

Tested in service on the final application, the proposed power source provides a stable output current. The subsequent constant glow is optimum to obtain conclusive spectroscopic measurements. It is possible to identify pollutant species in water when they are present in low concentrations.

**Author Contributions:** Conceptualization, P.J.V. and J.A.M.-R.; methodology P.J.V., D.G.C. and J.A.M.-R.; validation, P.J.V. and J.A.M.-E., D.B.F. and G.M.-R.; resources, P.J.V.; writing—original draft preparation, P.J.V. and J.A.M.-E.; writing—review and editing, P.J.V. and J.A.M.-E. All authors have read and agreed to the published version of the manuscript.

**Funding:** This research was funded by FUNDACION PARA LA INVESTIGACION CIENTIFICA Y TECNICA FICYT, grant number SV-PA-21-AYUD/2021/50938.

**Conflicts of Interest:** The authors declare no conflict of interest. The funders had no role in the design of the study; in the collection, analyses, or interpretation of data; in the writing of the manuscript; or in the decision to publish the results.

## Appendix A

Considering the circuit in Figure 12, two of Kirchhoff's laws can be easily deduced, as given in (A1), which explains (6):

1. The voltage in the inductor, $L_F$, is equal to the voltage in the resistor, $R_X$.
2. The voltage in the capacitor is equal to the addition of the voltages in the inductor, the resistor, $R_A$, and $V_A$. $R_A$ represents the negative resistor and $V_A$ represents the voltage of the arc in the glow discharge steady state.

These equations are applied when the capacitor is still charged at the breakdown voltage, but the gap is already behaving at glow discharge. The circuit experiments describe a transient as follows:

$$\left.\begin{aligned} V_{LF} &= V_{RX} \\ V_{C0} &= V_{LF} + V_{RA} + V_A \end{aligned}\right\} \rightarrow \left.\begin{aligned} L_F\frac{d\,i_L}{dt} &= R_X(I_0 - i_L) \\ V_F = \frac{1}{C_0}\int_{-\infty}^{t} -I_0 \cdot dt &= L_F\frac{d\,i_L}{dt} + R_A \cdot I_0 + V_A \end{aligned}\right\} \tag{A1}$$

The upper part of (A1) is manipulated to isolate $I_0$:

$$I_0 = i_L + \frac{L_F}{R_X} \cdot \frac{di_L}{dt} \tag{A2}$$

The derivative of (A2) gives:

$$\frac{dI_0}{dt} = \frac{di_L}{dt} + \frac{L_F}{R_X} \cdot \frac{d^2 i_L}{dt^2} \tag{A3}$$

Now, the derivative of the lower part of (A1) gives:

$$\frac{-I_0}{C_0} = L_F \frac{d^2 i_L}{dt^2} + R_A \cdot \frac{d I_0}{dt} \tag{A4}$$

Hence, (A2) and (A3) are used to modify (A4):

$$\frac{-i_L}{C_0} - \frac{L_F}{C_0 \cdot R_X} \cdot \frac{di_L}{dt} = L_F \frac{d^2 i_L}{dt^2} + R_A \cdot \frac{di_L}{dt} + \frac{R_A}{R_X} L_F \cdot \frac{d^2 i_L}{dt^2} \tag{A5}$$

Operations are made to re-express (A5) in a more compact manner, which explains (7):

$$\frac{d^2 i_L}{dt^2} L_F \left(1 + \frac{R_A}{R_X}\right) + \frac{di_L}{dt} \left(R_A + \frac{L_F}{C_0 \cdot R_X}\right) + i_L \frac{1}{C_0} = 0 \tag{A6}$$

$$\frac{d^2 i_L}{dt^2} + \frac{di_L}{dt} \left(\frac{R_A \cdot R_X}{R_X + R_A} \frac{1}{L_F} + \frac{1}{C_0 \cdot (R_X + R_A)}\right) + i_L \frac{1}{C_0 L_F} \cdot \frac{R_X}{R_X + R_A} = 0 \tag{A7}$$

The coefficients of (A6) can be organized according to their physical meaning as time constants. In this way, (A6)–(A2) justify (7) to (11):

$$\tau_L = L_F \left(1 + \frac{R_A}{R_X}\right) = L_F \left(\frac{R_A + R_X}{R_X \cdot R_A}\right) \tag{A8}$$

$$\tau_C = C_0 \cdot (R_X + R_A) \tag{A9}$$

The initial conditions are currently equal to zero in the inductor and breakdown voltage in the capacitor:

$$\left.\begin{array}{c} i_L(0) = 0 \\ V_F = V_{BD} = L_F \left.\frac{d i_L}{dt}\right|_0 + R_A \cdot I_0|_0 + V_A \end{array}\right\} \tag{A10}$$

Applying (A2) into (A9):

$$\left.\begin{array}{c} i_L(0) = 0 \\ V_{BD} = L_F \left.\frac{d i_L}{dt}\right|_0 + R_A \cdot i_L(0) + R_A \cdot \frac{L_F}{R_X} \cdot \left.\frac{d i_L}{dt}\right|_0 + V_A \end{array}\right\} \rightarrow \left.\begin{array}{c} i_L(0) = 0 \\ \left.\frac{d i_L}{dt}\right|_0 = \frac{V_{BD} - V_A}{L_F(R_X + R_A)} \end{array}\right\} \tag{A11}$$

Now, (A6), the equation of the system, can be solved with the aid of any mathematical application, aiding to the design of the filter.

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
