# Peer review of "High-Voltage LC-Parallel Resonant Converter with Current Control to Detect Metal Pollutants in Water through Glow-Discharge Plasma"

_electronics, doi:10.3390/electronics11040644_

Round 1

Reviewer 1 Report

This paper describes a power source for a detector. Overall, the logic is not clear enough, and the authors need to think about what the significance of this research is. More important, the authors should highlight the main issue addressed via the study in this article. Some comments are listed here:

  1. The title has no bright spots and cannot arouse the interest of readers.
  2. Abstract does not summarize the full text well.
  3. The content logic of introduction is not clear. It seems that some parts of the method are also written in the introduction section.

(What is the progress in the fields involved in this research? What are the advantages of this research compared with other research? etc. are not mentioned)

  1. The figures might be too much.
  2. The figures of data graphs or result graphs, are displayed in a very imprecise form. This makes the article appear to be of low quality.
  3. The discussion section are not clear enough. Also, the article does not have any in-depth discussion, which is one of the most important elements for a scientific paper.

Author Response

Answer to the reviewers:

Reviewer 1

  1. According to the good idea of the reviewer, the title has been reconsidered to include many of the theme keywords, trying to get a better impact in any internet search on the matter. In particular, the title contains now the following searching terms:

- High voltage

- Current control

- Resonant converter

- Parallel resonant converter.

- LC parallel resonant converter.

- Metal pollutants

- Water pollutants

- Glow discharge

- Plasma

2.- The abstract has been thoroughly rewritten to summarize better the idea of the paper, raising the issue and showing the novelties and the technical study it includes. In particular:

- It presents a single stage solution, based on a resonant converter capable to attain the very different operation conditions in the load.

- It describes how to design the power stage for the application, paying special attention to the transformer design, the corner stone in the converter.

- It models and explains the control loop and a new filter conception to make the feedback loop prone to stability, illustrating the idea in the frame of an application.

- It demonstrates the good behavior of the whole converter once connected to the load and provides information about the system performance.

The new text of the abstract is:

“This paper presents a high voltage power source to produce glow discharge plasma in the frame of a specific application. The load has two well-differentiated types of behavior. To start the sys-tem it is necessary to apply a high voltage, up to 15 kV, to produce the air dielectric breakdown. Before that the output current is zero. On the contrary, under steady state, the output voltage is quite smaller, a few hundreds of volts, while the load requires a current source behavior to maintain a constant glow in the plasma. The amount of current must be selectable by the opera-tor in the range 50 mA – 180 mA. Therefore, very different voltage gains are required, and they cannot be easily attained by a single power stage. This work describes why the LC parallel resonant topology is a good single stage alternative to solve the problem and shows how to make the design. The step up transformer is the key component of the converter. It provides galvanic isolation and adapts the voltage gain to the most favorable region of the LC topology but it also introduces non-avoidable reactive components for the resonant net, determining their shape and, to some extent, their magnitude. In the paper, transformer constructive details receive a special attention, discussing its model. The experimental dynamic tests, carried out to design the control, show a load behavior that resembles a negative resistance. This fact makes any control loop prone to instability. To compensate this effect, a resistive ballast is proposed, eliminating its impact on efficiency with a novel filter design, based on an inductor, connected in series with the load beyond the voltage clamping capacitor. The analysis includes a mathematical model of the filtering capacitor discharge through the inductor during the breakdown transient. The model provides insight to dimension the inductor, to limit discharge current peak and to analyze overall performance on steady state. Another detail addressed is the balance among total weight, efficiency and autonomy which appears if the filter inductor is substituted for a larger battery in autonomous operation.

Finally, a comprehensive set of experimental results on the real load illustrate the performance of the power source, showing waveforms at breakdown and at steady state (for different output currents). Additionally, the detector constructive principia are described and its experimental performance is explored, showing results with two different types of metallic pollutants in water”

3.- Following the recommendations of the reviewer, changes have been made in the introduction to state the novelties presented in the paper. The new rewritten text is:

“A very common approach in other plasma applications is the use of several stages in cascade [12 -13]. However, for this low power, portable application the final structure should be as simple as possible. Bearing that in mind, resonant converters have the capability to adapt to very different output conditions and they are widespread in industry for many applications [14-18]. In fact, a comprehensive review of power supplies for plasma materials processing [13] shows that a solution based on LLC or LCC resonant structures have already been proposed as intermediary power conversion systems [19-21]. In this paper, however, the LC parallel resonant topology is preferred. It presents a simpler resonant net and can integrate easily the step up transformer and its parasitic components. Moreover, the application requires a wide variation in the gain (3), which requires a power topology capable to deal inherently with very different conversion ratios. The LC solution presents such a flexibility to a greater degree than other resonant topologies and their features match those of the application: at no load resonance leads to high voltage gain, while voltage attenuation is possible at different output currents without a large excursion in switching frequency. All these facts make the LC resonant structure a better candidate for the application.

In the next paragraph, the model of the LC parallel resonant topology with a capacitor as output filter also known as PRC-C, Fig. 2, is used to carry out the design.”

  1. y 5.- Fig. 10c, 15c and 15d have been eliminated and figs. 6, 7, 8, 9, 10, 14 and 15 have been improved.

6.-  The conclusions have been rewritten to be more concise and assertive. Supported by facts and figures, they clearly state the contributions of the paper.

The new text of the conclusions is:

“This paper addresses an application where voltage gain varies from 9.4 to 312 when the input remains constant at 48 V. Such a ratio, larger than 1 to 33, is very difficult to attain by a single-stage converter. However, a LC parallel resonant topology has been demon-strated as a good solution, being capable to operate in the whole range with good perfor-mance. In fact, a successful design is explained, assembled and experimentally tested. The analysis shows that the step up transformer is the key component of the topology, and that it has to be optimized in many ways at the same time. In particular, it is concluded that:

1.- Resonant intrinsic voltage gain (transformer ratio excluded) must be centered in the topology optimum range, i.e., 0.3 to 7.5. condition transformer ratio.

2.- Transformer magnetizing inductance must be big enough not to affect the resonant net, keeping it simple. The design shows the value and the core size involved.

3.- Secondary winding can be wound in separate columns to simplify the assembly. A concentric secondary coil to ensure isolation in the available winding area is proposed.

4.- Leakage inductance can be maintained under control with that structure. Isolation reduces parallel capacitance, therefore it can be externally controlled to configure a convenient switching frequency.

Regarding the control, the paper shows how to obtain experimentally the low signal behavior of the load, demonstrating that it acts as a negative resistor. As such a load makes the control loop prone to instability, a new filter conception based on resistor and filtering inductor in series with the output has been introduced to easily compensate that. The mathematical analysis explains accurately the way it behaves dynamically and at steady state. It provides all the information required to dimension the filter, which has been illustrated and tested in the prototype. Additionally, considering the size, it is demonstrated that the filter efficiency is preferable to the same weight of extra battery to maintain autonomy. The main part of remaining losses fall on the output rectifying diodes. They pre-sent a very large forward voltage as penalty to withstand the breakdown output voltage. Further improvement in efficiency might be obtained from a new rectifying structure.

Finally, the final power source is tested in service on the final application. Output current stability provides a constant glow, optimum to obtain conclusive spectroscopic measurements. Pollutant species have been sharply detected by optical analysis in the equipment.”

Reviewer 2 Report

Dear Authors, It is a very interesting paper that deserves to be published after minor corrections.

Pages 5-6, lines 162 to 179 and lines 180 to 202: repetition of the text.

Page 7, line 232, equation 6 to justify from the corresponding equation.

Page 12, lines 345-346, equations 6, 8, 7 to justify from the corresponding equations.

Page 13, line 358, equation 7 to justify from the corresponding equation.

Figures 7-8-9-11: change the hardly visible yellow color on the curves.

Figures 9-10: remove the gray background for better visibility.

In reference 19, write bipolar.

In figures 7, 8, 9, 10, 11, 14 it is not possible to read the small text or symbol in the margin of the Y axes.

Author Response

Answer to the reviewers:

Reviewer 2

  1. Pages 5-6, lines 162 to 179 and lines 180 to 202: repetition of the text.

This error has already been corrected in the final text.

  1. Page 7, line 232, equation 6 to justify from the corresponding equation.

  1. Page 12, lines 345-346, equations 6, 8, 7 to justify from the corresponding equations.

  1. Page 13, line 358, equation 7 to justify from the corresponding equation.

Points 2, 3 and 4 have been justified in an annex at the end of the paper.

  1. Figures 7-8-9-11: change the hardly visible yellow color on the curves.

Figures 7, 8, 9 and 11 have been modified to make them more visible.

  1. Figures 9-10: remove the gray background for better visibility.

Figures 9 and 10 have had the gray background color removed to improve visibility.

  1. In reference 19, write bipolar.

This point has been corrected.

  1. In figures 7, 8, 9, 10, 11, 14 it is not possible to read the small text or symbol in the margin of the Y axes.

In figures 7, 8, 9, 10, 11, 14 the text or symbol has been corrected outside the Y axes for better viewing.

Round 2

Reviewer 1 Report

The revised manuscript has been improved, however, there is still much room for improvement. It is suggested that the authors make further revisions.

Specifically, there are two major recommendations:

  1. The discussion section is not deep enough, and it is recommended to strengthen the discussion. This is also the problem mentioned in the previous review comments (No. 6).

  1. The conclusion is too long, please rewrite it.

Author Response

Answer to the reviewers:

Reviewer 1

  1. Following your recomendations, we have added a new section for discussion. There, we have addressed the main topics of the paper, reflecting on the work done, assesing the results and putting them into perspective.

“In a low power application, below 200W, a single stage power source is preferable. In this case, gain conditions introduce an additional difficulty since, with a wide variation ratio from 1 to 33, they are very difficult to attain. The LC parallel resonant topology has been demonstrated as a good solution to cope with both conditions. It provides more flexibility than other resonant structures as LLC or LCC, which have been previously proposed as intermediate stages. In fact, the parallel resonance presents the largest gain for a given quality factor, while gains well below one are still possible within a short switching fre-quency excursion. Additionally, LC structure adapts the parasitic components of the step up transformer in a natural way. Regarding control, a new filter structure stabilizes over-all behavior without any efficiency penalty. However, the need to use four high voltage diodes in the output rectifier remains. Their large forward voltage deteriorates efficiency in at least 5% when the output voltage is low at glowing discharge. Further improvement in efficiency might be obtained from a new rectifying structure. Overall, the final design re-mains simple, robust and functional.”

  1. The conclusions have also been rewritten and adpated to the new scheme of the paper.

“This paper demonstrates that the LC parallel resonant topology is a good candidate for a power source to maintain glowing discharge in air at atmospheric conditions. The keys of the design rely on:

1.- An intrinsic voltage gain (without step up transformer) centered in the stage optimum range: 0.3 – 7.5.

2.- A step up transformer with a magnetizing inductor that can be ignored in any resonant net and does not affect the core size. A standard size core that supports windings in sepa-rate columns with enough isolation and provides a suitable leakage inductance.

The experimental low signal model of the load shows that its dynamic behavior is prone to instability. A new filter, based on a resistor and a filtering inductor compensates the equivalent negative resistor of the system, not affecting the efficiency. A mathematical model of such a filter provides a guide to the design which has been applied and experi-mentally demonstrated in the application. Final weight of the filter is preferable to the equivalent battery size to compensate a worse performance. Most of remaining losses falls on the output rectifying diodes: They present a very large forward voltage as penalty to withstand the breakdown output voltage.

Tested in service on the final application, the proposed power source provides a stable output current. The subsequent constant glow is optimum to obtain conclusive spectro-scopic measurements. It has been possible to identify pollutant species in water when they are present in low concentrations.”

Round 3

Reviewer 1 Report

The manuscript has been revised based on previous comments. I think this paper can be accepted.